# Emerging Role of the Inflammasome and Pyroptosis in Hypertension

**DOI:** 10.3390/ijms22031064

**Published:** 2021-01-21

**Authors:** Carmen De Miguel, Pablo Pelegrín, Alberto Baroja-Mazo, Santiago Cuevas

**Affiliations:** 1Section of Cardio-Renal Physiology and Medicine, Division of Nephrology, Department of Medicine, University of Alabama at Birmingham, Birmingham, AL 35233, USA; 2Molecular Inflammation Group, Biomedical Research Institute of Murcia (IMIB-Arrixaca), 30120 Murcia, Spain; pablo.pelegrin@imib.es (P.P.); alberto.baroja@ffis.es (A.B.-M.)

**Keywords:** hypertension, pyroptosis, inflammasome, inflammation

## Abstract

Inflammasomes are components of the innate immune response that have recently emerged as crucial controllers of tissue homeostasis. In particular, the nucleotide-binding domain, leucine-rich-containing (NLR) family pyrin domain containing 3 (NLRP3) inflammasome is a complex platform involved in the activation of caspase-1 and the maturation of interleukin (IL)-1β and IL-18, which are mainly released via pyroptosis. Pyroptosis is a caspase-1-dependent type of cell death that is mediated by the cleavage of gasdermin D and the subsequent formation of structurally stable pores in the cell membrane. Through these pores formed by gasdermin proteins cytosolic contents are released into the extracellular space and act as damage-associated molecular patterns, which are pro-inflammatory signals. Inflammation is a main contributor to the development of hypertension and it also is known to stimulate fibrosis and end-organ damage. Patients with essential hypertension and animal models of hypertension exhibit elevated levels of circulating IL-1β. Downregulation of the expression of key components of the NLRP3 inflammasome delays the development of hypertension and pharmacological inhibition of this inflammasome leads to reduced blood pressure in animal models and humans. Although the relationship between pyroptosis and hypertension is not well established yet, pyroptosis has been associated with renal and cardiovascular diseases, instances where high blood pressure is a critical risk factor. In this review, we summarize the recent literature addressing the role of pyroptosis and the inflammasome in the development of hypertension and discuss the potential use of approaches targeting this pathway as future anti-hypertensive strategies.

## 1. Inflammation and Hypertension

Elevated blood pressure, defined as a systolic pressure higher than 130 mmHg and diastolic pressure greater than 80 mmHg [1], is the leading risk for cardiovascular and kidney diseases [2] and was reported to affect 1.13 billion people in the world in 2016 [3]. As a consequence, high blood pressure is a leading cause of death worldwide [4] and research in the field of hypertension is highly dynamic. Extensive evidence demonstrates an important role of inflammation in the pathogenesis of hypertension and vascular and kidney diseases. A series of elegant studies performed by Grollman et al. in the 1960s first evidenced that autoimmune factors play a critical role in an animal model of renal infarction-induced hypertension [5,6]. The same group also reported that hypertension could be transferred to normotensive rats by transplanting lymph node cells from rats with renal infarction hypertension [7]. In the 1970s, Svendsen revealed that an intact thymus is required for the maintenance or development of hypertension in three different animal models of the disease: the deoxycorticosterone acetate (DOCA) salt model [8], mice with partially infarcted kidneys [9], and the genetically hypertensive NZB/Cr mouse strain [10].

Since those early studies, investigation of the interplay between inflammation and high blood pressure has grown exponentially, and more than 18,000 publications have explored this topic to date. It is now widely acknowledged that low-grade or persistent inflammation is a key player in the development and maintenance of hypertension. Exhaustive research demonstrates the infiltration of immune cells, like T cells, macrophages, and dendritic cells in the kidneys, perivascular fat, or heart during the development and progression of hypertension [11,12,13,14,15,16,17,18,19,20,21,22,23,24]. In particular, immune cell transfer studies performed by Guzik et al. [15] demonstrated that the development of angiotensin II-induced and DOCA salt-induced hypertension was dependent on the presence of T cells. Moreover, pharmacological inhibition of immune cells with, for instance, mycophenolate mofetil, tacrolimus, or cyclophosphamide attenuates the development of high blood pressure [25,26,27,28,29]. More recently, the genetic ablation of certain immune cells and specific receptors on immune cells or cytokines demonstrates that activation of immune cells like T cells, macrophages, and dendritic cells [18,23,30,31,32,33,34,35] is essential for the development of elevated blood pressure. Lately, B cells have also been reported to be important in the development of hypertension [36]. The role of immune cells from the adaptive and innate arms of the immune response as well as different cytokines in hypertension has been extensively reviewed elsewhere, and thus, we will not review it here [37,38,39,40,41,42]. 

As mentioned above, inflammatory cells have been repeatedly demonstrated to infiltrate organs involved with blood pressure regulation, including the vasculature, the kidneys, and the brain [40,43,44], during hypertension. For instance, in a hypertensive kidney, macrophages and T cells localize around glomeruli and arterioles and within the interstitium [45,46]. Similarly, infiltration of immune cells and increased levels of inflammatory mediators have been demonstrated in the perivascular fat of large arteries and arterioles in animal models of hypertension [45]. Activated inflammatory cells produce and release cytokines like tumor necrosis factor alpha (TNF-α), interleukin-1 (IL-1), IL-17, IL-6, and interferon-gamma (IFN-γ) that are known inducers of renal and vascular dysfunction [33,34,35,47,48,49]. Many of those inflammatory mediators are known to stimulate fibrosis [47,50,51], for instance, which is seen in both vessels and kidneys of hypertensive patients and animal models of the disease. In addition, T cells and macrophages contain all the machinery needed for the production of vasoactive molecules such as angiotensin II [52,53,54], endothelin-1 [55,56], or prostaglandins [57,58], which are known mediators of hypertension and hypertension-induced end-organ damage. Moreover, a number of inflammatory factors released by activated immune cells can also modulate the local production of angiotensinogen and, later, angiotensin II generation within the kidneys, vasculature, or nervous system [51,59,60]. This implies that inflammatory cells increase the local levels of pro-hypertensive stimuli and can further increase blood pressure by stimulating fluid retention and vascular constriction [61]. Interesting animal studies have also shown that IL-17 or IFN-γ deficiency is associated with alterations of sodium (Na^+^) transporters in the kidneys and, consequently, reduced Na^+^ retention in hypertensive conditions [62]. Despite all this evidence linking immune cell infiltration and the development and progression of hypertension, the exact mechanisms that trigger low-level inflammation observed in the elevated blood pressure setting still remain unclear.

## 2. Inflammasomes and Pyroptotic Cell Death

Inflammasomes are essential players in the inflammatory response and one of the first steps for the initiation of chronic low-grade inflammation. Inflammasomes are intracellular protein oligomers that are sensors for pathogens, tissue injuries, and recognition of the signals altering homeostasis. They are composed of an effector protein, an adaptor protein, and a sensor protein that oligomerize in a large complex after activation. The inflammasome sensor protein is usually a pattern recognition receptor (PRR) that oligomerizes after activation induced by pathogen-associated molecular patterns (PAMPs), endogenous host-derived damage-associated molecular patterns (DAMPs), or homeostasis-altering molecular processes (HAMPs) [63]. Three families of sensor proteins have been described, belonging to the nucleotide-binding domain, leucine-rich-containing family of receptors (NLRs), the absent in melanoma 2-like receptors (ALRs) or pyrin, which give name to the inflammasome and share analogous structural domains. Moreover, there are several types of inflammasomes, such as NLRP1b (mouse), NLRP3, NLR family CARD domain-containing protein 4 (NLRC4), NLRP6, NLRP9, pyrin, and absent in melanoma (AIM) 2. A great variety of molecules activate different sensor proteins, including *Bacillus anthracis* for NLRP1b, *Salmonella typhimurium* for NLRC4, lipoteichoic acid for NLRP6, viral dsRNA for NLRP9, dsDNA for AIM2, toxin-induced modifications of Rho GTPases for pyrin, and several PAMPs, DAMPs, and HAMPs for NLRP3 [64]. After activation, the sensor protein oligomerizes and recruits the adaptor protein apoptosis-associated speck-like protein with the caspase recruitment domain (ASC) forming large filaments by prion-like oligomerization [65]. ASC filaments recruit the effector zymogen pro-caspase-1 by caspase activation and recruitment domain (CARD)–CARD homotypic interaction, facilitating its autoactivation by close proximity [66]. However, other inflammasomes can be activated without involvement of ASC, such as NLRC4 [67]. Activated caspase-1 induces the cleavage of gasdermin D and releases its lytic N-terminal domain (GSDMD^NT^) [68,69] from its C-terminus repressor domain to form pores in membranes. GSDMD^NT^ lyses mammalian cells by acting from the inside of the cells and also has anti-bacterial activity [70]. Studies of the full-length crystal structure of GSDMD reveals distinct features of auto-inhibition among the gasdermin (GSDM) family members [71]. Released GSDM^NT^ domains bind to negatively charged membrane lipids like phosphatidylinositol phosphate, cardiolipin, and phosphatidylserine [72]. Caspase-1 activation also mediates the proteolytic cleavage of the inactive precursor cytokines pro-IL-1β and pro-IL-18 to produce their active forms that are mainly released by GSDMD pores, which have a diameter of 10–14 nm and are big enough to release the mature form of these cytokines [70] (Figure 1). The GSDMD pore consists of about 54 N-terminal gasdermin subunits with a radical conformational change compared to their conformation in the full GSDM sequence, with extensive interactions among the inserted β-strands and the α-helix 1 localized in the globular domain [73]. Upon permeabilization of the plasma membrane by GSDMD pores, cells undergo a lytic, pro-inflammatory cell death known as pyroptosis that further promotes and increases the release of mature IL-1β and IL-18 and the induction of a pro-inflammatory environment [74]. If gasdermin pores at the plasma membrane are not repaired, pyroptosis will end with a burst in pro-inflammatory cytokine release and the discharge of large intracellular components (such as inflammasome oligomers or high mobility group box 1 (HMGB1) among others), resulting in a highly pro-inflammatory cell death process [68]. Unlike other programmed cell deaths, pyroptosis has been described as a strategy to preserve the inflammatory response through the release of pro-inflammatory intracellular content during plasma membrane permeabilization [75,76,77], a process that recruits immune cells to combat invading infectious organisms and promote tissue healing [78]. Pyroptosis was initially reported in macrophages infected with *Salmonella typhimurium* [79], but later it was also demonstrated in other cell types and in response to different signals [80]. Since then, pyroptosis has been associated with multiple diseases such as cardiovascular, liver, kidney, inflammatory, and immune diseases or cancer [81]. 

In humans, GSDM family members are composed of 6 members: GSDMA, B, C, D, E, and Pejvakin (PJVK), and all have a highly conserved N-terminal domain that induces pyroptosis when expressed ectopically, except for PJVK^NT^ [68]. Pyroptosis depends on different innate immune pathways leading to GSDMD processing. Caspase-1 is one of the most studied proteases involved in pyroptosis by GSDMD cleavage. However, GSDMD cleavage and pyroptosis are also executed by other inflammatory caspases: human caspase-4 and -5 and mouse caspase-11 [80,82]. Moreover, other caspases (like apoptotic caspase-3 or -8) and proteases (like elastase or granzyme) can also elicit processing of GSDMB, D, and E and induce pyroptosis [83].

NLRP3 is the most studied member of the inflammasome family as it has been implicated in the pathophysiology of several autoinflammatory syndromes [84] and other diseases associated with metabolic, degenerative, and inflammatory processes [85]. Activation of the NLRP3 inflammasome can be induced in two different manners: the “canonical” and “non-canonical” pathways. 

Canonical NLRP3 inflammasome activation begins with a priming signal driven by several classes of receptors facilitating the upregulation of NLRP3 and pro-IL-1β via nuclear factor kappa B (NF-κB) signaling and also key post-translational modifications of NLRP3. An activation signal is then initiated by PAMPs, DAMPs, or HAMPs, inducing physiological changes that are detected by NLRP3 and followed by the recruitment of ASC and caspase-1 to form the inflammasome complex [86]. Among the activators of this cascade, our group found different microorganisms, bacterial pore-forming toxins, hemozoin, or even melittin [87,88]. NLRP3 can also be activated by the phagocytosis of particulate matter (silica, alum, asbestos, uric acid crystals, cholesterol crystals, or β-amyloid deposits) as well as by cell swelling [89] or extracellular adenosine triphosphate (ATP) acting through the P2X purinoceptor 7 (P2X7) [90]. This activation signal islinked to mitochondrial dysfunction, lysosomal destabilization or plasma membrane damage, which result in cell metabolic changes, reactive oxygen species (ROS) formation, potassium (K^+^) and chloride (Cl^–^) efflux, and calcium (Ca^2+^) influx [91]. Although the exact mechanism leading to NLRP3 activation is not well-known, K^+^ efflux is a common initial step for NLRP3 activation in response to most of the triggers [92,93]. 

In the non-canonical inflammasome pathway, gram-negative bacteria activate the toll-like receptor 4 (TLR4)–toll IL-1 receptor (TIR)-domain-containing adapter-inducing interferon (IFN)-β (TRIF) signaling pathway leading to type I IFN production and upregulation of guanylate-binding proteins (GBPs) and immunity-related GTPase family member B10 (IRGB10), which in turn target outer membrane vesicles (OMVs) or bacterial and vacuolar membranes to facilitate the release of lipopolysaccharides (LPS) into the cytoplasm [94]. LPS can directly interact with murine caspase-11 or human caspase-4/5 and activate NLRP3 as a consequence of K^+^ efflux induced by GSDMD pores in the cell membrane or after sensing DAMPs released by pyroptotic cells, suggesting a functional crosstalk between the canonical and non-canonical pathways [95]. 

On the other hand, GSDMD pore formation has been found responsible for IL-1β release in the absence of pyroptotic cell death [96]. Likewise, calcium influx through GSDMD pores can act as a signal to induce membrane repair by recruiting the endosomal sorting complexes required for transport (ESCRT) machinery to damaged plasma membrane areas, inducing cell survival mechanisms beyond pyroptosis [97]. Moreover, IL-1β release upon activation of caspase-1 can be independent of GSDMD membrane permeabilization [98]. The different mechanisms involved in the activation of the NLRP3 inflammasome is a topic of active investigation and thus, in addition to the mechanisms described above, alternative molecular mechanisms have also been unveiled recently. Because of space concerns, we only reviewed here the more established mechanisms, but we refer the reader to [99,100,101] for a more detailed description of these novel mechanisms.

## 3. Novel Roles of the NLRP3 Inflammasome and Pyroptosis in Blood Pressure Regulation

Fascinating new literature suggests an involvement of the inflammasome in the development and progression of hypertension along with the induction of end-organ damage. Immune cell activation is dependent on the presence of pattern recognition receptors (PRRs) on antigen-presenting cells. These receptors not only recognize molecules of pathogens or PAMPs, but they can also be activated by binding to DAMPs [41] that may be released by damaged cells in an organ injured during hypertension. Elevated activation levels of the key NLRP3 inflammasome priming factor, NF-κB, in tissue and inflammatory cells are persistent in hypertension [102,103] and result in exaggerated levels of circulating and tissue IL-1β and IL-18 in patients with essential hypertension and in animal models of the disease [104,105]. Polymorphisms in the *NLRP3* gene, specifically the rs7512998 variant, are associated with elevated blood pressure in older populations [106]. Those individuals with polymorphisms in the *NLRP3* gene develop higher elevations in blood pressure after the age of 50 than individuals without that gene variant [106]. A different polymorphism, in this case, an intronic 42 base pair variable number of tandem repeat (VNTR) in the *CIAS1* gene encoding NLRP3, has also been linked to susceptibility to develop essential hypertension [107]. 

### 3.1. NLRP3 Inflammasome in Human Pulmonary Hypertension

The involvement of the NLRP3 inflammasome in human pulmonary hypertension is also a topic of active investigation, as reviewed by Scott et al. [108], although research in this area is still in the early stages. In the pulmonary disease setting, expression of components of the inflammasome has been described in macrophages [109], endothelial cells [110,111], and lung epithelial cells [112]. In addition, lung-specific overexpression of IL-18 in mice was also shown to associate with pulmonary hypertension [113], while IL-18 suppression attenuates that blood pressure elevation [114] and IL-18 receptor deficiency results in protection against a respiratory insult such as cigarette smoke [115]. Taken together, this evidence highlights the importance of the inflammasome pathway in inducing lung disease. Moreover, absence of the ASC adaptor protein attenuates hypoxia-induced pulmonary hypertension in mice [116]. Pre-clinical and clinical studies further support the involvement of the NLRP3 inflammasome in pulmonary hypertension [117,118,119,120,121,122,123,124] and confirm the occurrence of pyroptosis in rat pulmonary arteries and hypoxic human pulmonary smooth muscle cells [125]. Although more research is clearly needed to better define the exact mechanisms mediated by the NLRP3 inflammasome in leading to pulmonary hypertension, the current evidence surely hints to this pathway as a presumed therapeutic target in this devastating disease.

### 3.2. NLRP3 Inflammasome in Pre-Eclampsia and Systemic Hypertension

Studies have also reported activation of the NLRP3 inflammasome in monocytes during pre-eclampsia [126], demonstrating a crucial role of NLRP3 inflammasome activation in increasing maternal blood pressure during pregnancy. Similarly, monocytes obtained from hypertensive patients demonstrate exaggerated production of IL-1β when isolated and stimulated in vitro with known hypertensive stimuli like angiotensin II [127]. Importantly, a recent unbiased next-generation RNA sequencing approach revealed the differential expression of 60 genes between monocytes isolated from normotensive and hypertensive individuals [128]. Most of those genes were involved in the IL-1β pathway [128], thus implying an overstimulation of the inflammasome pathway in these cells during hypertension. These same investigators also identified a correlation between expression of the IL-18 receptor accessory protein (IL-18RAP) and elevated blood pressure in a cohort of African American patients and suggested this protein as a possible novel immune target in hypertension [128]. 

In animal studies, activation of the NLRP3 inflammasome has been demonstrated in cardiovascular organs and those important for blood pressure regulation like the kidneys, the vascular endothelium, or the hypothalamus, organs that are often affected by persistent elevations of blood pressure. For instance, in the mouse model of DOCA salt-induced hypertension, an increased expression of components of the NLRP3 inflammasome was described in the kidneys, as well as elevated levels of IL-1β [129]. Most interestingly, the elevation of blood pressure in response to DOCA salt could be prevented by using MCC950, a novel NLRP3 inflammasome inhibitor that prevents its oligomerization and activation [129]. Furthermore, in very recent studies, this same research group demonstrated that inhibition of the NLRP3 inflammasome was also effective in reducing the renal dysfunction, inflammation, and fibrosis associated with DOCA salt-induced hypertension [130]. Supporting the critical role of the NLRP3 inflammasome in the development of hypertension, genetic interventions to knockout NLRP3 or ASC proved to prevent the elevation of blood pressure observed in the mouse two kidney, one clip (2K1C) model of hypertension [131]. Others demonstrated that endothelial nitric oxide synthase (eNOS)-deficient mice exhibit exaggerated renal NLRP3 inflammasome activation after an aldosterone-induced renal damage protocol [132], emphasizing that the eNOS–NO pathway controls the activation of the NLRP3 inflammasome and keeps it in check. This is especially significant because a deficient eNOS function has been repeatedly demonstrated in hypertension [133,134,135]. Another recent study used a rat model of hypertension involving NOS inhibition with L-N^G^-nitroarginine methyl ester (L-NAME) to show that blockade of NF-κB attenuates blood pressure and protects against hypertension-induced kidney damage, underscoring the importance of inflammasome priming in hypertension [136]. Other inflammasomes, such as the NLRC4 inflammasome, have been tentatively implicated in blood pressure regulation in mice [137]; however, further studies are needed to confirm this involvement. 

### 3.3. Renal Inflammasome Activation and Pyroptosis in Hypertension

Despite the evidence reviewed above describing activation of the NLRP3 inflammasome in the kidneys during hypertension, there is a lack of information in the literature regarding the presence of renal pyroptosis in the hypertensive setting. A recent small pilot study performed in Germany investigated the levels of inflammatory cell pyroptosis in hemodialysis patients and hypertensive patients with intact renal function and found that hypertensive patients had exaggerated active caspase-1 expression in monocytes compared to hemodialysis patients [138]. These findings suggest that the dialysis protocol was effective in removing pyroptotic cells from the circulation and confirm the outcomes described by others of elevated activation of the inflammasome pathway in monocytes isolated from hypertensive patients [127]. Increasing evidence, however, demonstrates exaggerated activation of the inflammasome in renal disease. For instance, activation of the NLRP3 inflammasome in glomerular cells and podocytes was associated with the progression of renal damage in the diabetic setting and in HIV-associated nephropathy [139,140]. Similarly, silencing of GSDMD-mediated pyroptosis in glomerular cells inhibits cell death and protects renal function in hyperglycemic conditions [141]. Deposition of the calcium carbonate crystal, which is crucial in the mechanism of tubular injury and kidney fibrosis during kidney stone pathology, has also been shown to stimulate the NLRP3 inflammasome via the transforming growth factor beta receptor (TGFR) signaling pathway [142]. Other reports suggest that the inflammasome cascade might considerably contribute to glomerular damage. All the components of the inflammasome are present in podocytes and its activation in these cells contributes to glomerulosclerosis in a model of hyperhomocysteinemia [143]. NADPH oxidase and ROS [144,145] have also been identified as potential triggers of the NLRP3 inflammasome in podocytes. This is highly significant because increased generation of reactive oxygen species is widely known to be intimately related to the development of hypertension [146,147]. Enhanced NLRP3 expression was detected in murine cultured podocytes, and human kidneys with mild signs of diabetic nephropathy exhibited immunohistochemical staining for NLRP3 in proximal and distal tubules as well as in sporadic cells that appeared to be podocytes [148]. Moreover, the inhibition of the NLRP3 inflammasome ameliorates renal injury in a variety of animal models [149]. Taken together, all this accumulating evidence for the expression and activation of the NLRP3 inflammasome in different kidney cell types during kidney diseases strongly suggests a role for pyroptosis in hypertensive kidney disease. Further in-depth investigation of the pyroptotic pathways in hypertensive kidneys is needed for the development of novel therapeutics that could prevent the progression of kidney disease in hypertensive patients.

### 3.4. Vascular Inflammasome and Pyroptosis Activation in Hypertension

Likewise, activation of the inflammasome pathway is also involved in the vascular dysfunction that occurs in hypertensive conditions. Evidence demonstrates that the NLRP3 inflammasome modulates the phenotype and proliferation rate of vascular smooth muscle cells in hypertension [150] and that the activation of the NLRP3 inflammasome in these conditions is triggered by increased cytosolic Ca^2+^ mediated by the calcium-sensing receptor CaSR [151]. The Ca^2+^-induced activation of the inflammasome elicits aortic fibrosis during hypertension [151]. It is also known that NLRP3-dependent pyroptosis in endothelial cells mediates endothelial dysfunction, which is one of the greatest contributors to the pathogenesis of hypertension [152]. Intriguingly, in vitro studies identified pyroptosis mediated by the NLRP3 inflammasome as a potential mediator of endothelial cell death after treatment with pollutants like cadmium or acrolein [153,154]. Other reports in the literature underline the possible role of noncoding RNAs, which are emerging biomarkers in the hypertension field [155], as regulators of endothelial function via the NLRP3 inflammasome signaling pathway [156]. Additionally, certain drugs that are often used to treat hypertension such as statins [157], hypoglycemic agents [158], and other anti-inflammatory drugs [159], together with specific inhibitors of NLRP3 such as MCC950 [160], have been reported to improve vascular dysfunction by inhibiting the NLRP3 inflammasome signaling cascade. Moreover, deletion of the NLRP3 gene attenuates the vascular remodeling and vascular smooth muscle cell transformation that is normally promoted by angiotensin II [161]. Interestingly, an inflammasome-independent role of NLRP3 has also been reported in the endothelial dysfunction that is induced by high salt consumption [162]. In summary, although growing evidence highlights the NLRP3 inflammasome and the consequent pyroptosis as important mediators of vascular dysfunction and vascular damage in hypertension, the exact molecular mechanisms that lead to activation of these damage pathways in this disease need more investigation.

### 3.5. Hypothalamic Activation of the Inflammasome During Hypertension

Fewer studies have focused on the activation of the inflammasome signaling cascade in blood pressure-controlling areas of the brain, like the hypothalamus. Chronic inhibition of NF-κB activity in the hypothalamic paraventricular nucleus (PVN) has been proven to delay the development of hypertension by upregulating anti-inflammatory cytokines, reducing NLRP3 and IL-1β expression and attenuating the enzymatic activity of the activated inhibitor of nuclear factor NF-κB kinase β (p-IKKβ, NF-κB p65) and NAD(P)H oxidase in the PVN of salt-sensitive hypertensive rats [163]. These findings suggest that high-salt-induced NF-κB activation in the PVN evokes hypertension via sympathoexcitation, which is related to the increases of NLRP3, IL-1β, and oxidative stress in the PVN [163]. Likewise, Wang et al. [164] found that NLRP3 expression in the PVN was significantly increased in a pre-hypertensive rat model and was accompanied by increased expression of pro-inflammatory cytokines, C-C motif ligand 2 (CCL2), CXC chemokine receptor type 3 (CXCR3), and vascular cell adhesion molecule 1 (VCAM-1). When the NLRP3 inflammasome was inhibited, it resulted in significantly decreased blood pressure and reduced pro-inflammatory cytokines, CCL2 and VCAM-1 [164]. On the other hand, a robust correlation between hypertension, depression, and other anxiety disorders has been identified [165,166]. It was reported that a strong association between chronic stress and depression exists and results in hippocampal and pre-frontal cortex atrophy by modulating the activity of the hypothalamic–pituitary–adrenal (HPA) axis and eliciting an oxidative stress and neuroinflammatory response [167]. If these effects in the morphology of the brain are related to hypertension is still under investigation; however, treatment with salvianolic acid B abolishes the chronic mild stress-induced depression through downregulating NLRP3 protein expression and results in improved behavior, antioxidant status, and anti-inflammatory activity [168]. The effects of modulation of the NLRP3 inflammasome pathway in the hypothalamus in a wider variety of animal models of hypertension are needed to achieve a deeper understanding of its role in the control of systemic blood pressure.

## 4. Targeting the NLRP3 Inflammasome and Pyroptosis in Hypertension: Emerging Pharmacological Approaches

Numerous antihypertensive drugs with anti-inflammatory properties such as cyclooxygenase (COX) inhibitors, angiotensin receptor blockers, or peroxisome proliferator-activated receptors (PPAR) gamma activators induce their anti-inflammatory effects indirectly by the inhibition of pro-inflammatory and pro-oxidant pathways [169]. Paradoxically, other anti-inflammatory agents, such as non-steroidal drugs and even some COX inhibitors, induce the opposite effect and increase blood pressure [170,171], indicating that the relationship between inflammation and blood pressure regulation is extraordinarily complex. 

As we described earlier in this review, numerous pieces of evidence demonstrate that the inflammasome and pyroptosis play a crucial role in essential and high-salt-dependent hypertension, as well as in preeclampsia, pulmonary hypertension, and in its related secondary disorders, evidencing that the attenuation of the inflammasome activity may be a promising target for achieving blood pressure control. The inflammasome is a key component in the regulation of the immune system and represents an initial step in the immune response. Several checkpoints in the inflammasome pathway have been tested to determine their efficiency in mitigating inflammasome activity and, thus, whether they constitute a possible target to reduce elevated blood pressure (Figure 2 and Table 1). In this section, we will highlight several of these checkpoints.

### 4.1. P2X7 Receptors Antagonism

P2X purinoceptors 7 (P2X7) are members of the ionotropic ATP-gated receptor family [195], and are involved in NLRP3 inflammasome activation in different diseases like depression and diabetes [196]. Recently, the selective P2X7 receptor antagonist A-740003 was reported to significantly attenuate the NLRP3 inflammasome upregulation and decrease the mean right ventricular (RV) pressure and RV hypertrophy associated with pulmonary hypertension [123]. On the contrary, treatment with a different P2X7 receptor antagonist, PKT100, did not alter right ventricular systolic pressure, but substantially improved survival in a mouse model of pulmonary hypertension [172], suggesting that further studies are needed to determine the possible role of P2X7 receptor antagonists in regulating blood pressure. 

### 4.2. Reactive Oxygen Species (ROS) Production Inhibitors

The phenolic compound ellagic acid ameliorates monocrotaline-induced pulmonary artery hypertension in rats via inhibiting the NLRP3 inflammasome signaling pathway, mostly due to its anti-oxidative properties [121]. Moreover, a recent study reports that ELABELA, a 32-residue hormone peptide, mitigates hypertension by attenuating the NADPH oxidase/ROS/NLRP3 inflammasome axis in human renal tubular cells stimulated with aldosterone, thus demonstrating the role of ROS in NLRP3 activation and blood pressure regulation in the kidneys [193].

### 4.3. NLRP3 Inhibitors

MCC950 is a specific NLRP3 blocker [160,197,198] and, as we mentioned before, it has been reported to partially reverse salt-induced hypertension in mice [130]. MCC950 reduces the renal expression of NLRP3 inflammasome subunits (NLRP3, ASC, pro-caspase-1) as well as inflammatory and injury markers (such as pro-IL-18, pro-IL-1β, IL-17A, TNF-α, osteopontin, ICAM-1, VCAM-1, CCL2, or vimentin) in hypertensive mice [84,130], while none of these parameters were altered by the same treatment in normotensive mice. These observations indicate that MCC950’s capacity to reduce blood pressure in salt-induced hypertension is due to the attenuation of the inflammasome activity [130]. In another study, Sprague–Dawley rats fed a high-salt diet were also treated with MCC950 via bilateral cannulae implanted into the hypothalamic paraventricular nucleus (PVN) for four weeks. At the end of the study, these rats presented with significantly increased NLRP3 expression in the PVN accompanied by increased microglia. The inhibition of the NLRP3 inflammasome significantly decreased blood pressure, reduced the expression of the chemokine CCL2, chemokine receptor CXCR3, and vascular cell adhesion molecule 1 (VCAM-1) and restored plasma norepinephrine (NE) expression levels [164]. Moreover, treatment with MCC950 was efficient in normalizing vascular ROS generation and reducing vascular dysfunction in db/db mice [191], and it also prevented the development of aortic aneurysms and dissections in mice [192]. Together, these data demonstrate the ability of MCC950 to reduce blood pressure by acting at both the hypothalamic level as well as the vascular level, improving endothelial function. 

### 4.4. NF-κB Inhibitors

As mentioned above, priming of the NLRP3 inflammasome is predetermined by the activation of NF-κB (Figure 1). Treatment with IMD-0354, which inhibits NF-kB activity by blocking the translocation of p65, effectively prevented the increase in right ventricular pressure and suppressed the proliferation and induced apoptosis of pulmonary arterial smooth muscle cells [194]. Further, infusion of pyrrolidine dithiocarbamate (PDTC), a different compound that also inhibits NF-κB, into the hypothalamic PVN hinders the development of high-salt-induced hypertension through the NLRP3 inflammasome and caspase-1 pathway [163], highlighting the possible relevance of preventing the development of hypertension by inhibiting NF-κB and caspase-1 activity at hypothalamic levels. 

### 4.5. IL-1β Receptor Antagonism

IL-1β and IL-18 are associated with hypertension [104,105] and hypertensive patients exhibit enhanced levels of IL-1β [138]. IL-1β stimulates the expression of vascular cell adhesion protein-1 (VCAM-1), intercellular adhesion molecule 1 (ICAM-1), and E-selectin in essential hypertension patients, which, in turn, results in unwanted atherosclerotic effects [199]. Enalapril, quinapril, and losartan are angiotensin-converting enzyme (ACE) inhibitors and angiotensin receptor blockers (ARBs) and are widely used as antihypertensive drugs. Use of these ACE inhibitors in hypertensive rats led to significantly decreased LPS-stimulated TNF-α and IL-1β levels, highlighting the ability of these drugs to attenuate inflammation [200]. Different studies revealed that polymorphisms in the IL-1β gene are associated with higher blood pressure in ethnic populations [165,201,202]; however, the question still remains whether IL-1β and IL-18 are inflammatory markers or mediators of hypertension in humans [203]. Serum levels of IL-1β and IL-1RA were described as predictors of elevated diastolic blood pressure in 537 subjects with insulin resistance syndrome [204]. In another study, incubation of the carotid arteries with LPS induced a greater concentration-dependent expression of mRNA for IL-1β in the spontaneous hypertensive rat (SHR) than in the normotensive Wistar–Kyoto rat [205]. Moreover, a study in anesthetized rats demonstrated that the intracisternal injections of IL-1β caused dose-dependent vasopressor responses; however, IL-1 did not constrict the peripheral vasculature in the perfused hindlimb preparation, which suggests that IL-1β may cause vasopressor effects via increases in the abdominal sympathetic discharge [206]. Anakinra is a specific recombinant antagonist of the IL-1β receptor currently approved by the U.S. Food and Drug Administration for use in inflammatory syndromes such as rheumatoid arthritis [173]. Preclinical studies in mice evidenced that Anakinra reduces blood pressure and renal fibrosis in the one kidney/DOCA salt-induced hypertension model [174]. In a phase II pilot study, Anakinra reduced pulmonary blood pressure and right ventricular failure [175]. However, a recent clinical study with a total of 9,549 participants reported that canakinumab, a specific anti-IL-1β blocking antibody, did not reduce blood pressure in hypertensive patients despite reducing major adverse cardiovascular event rates [176]. Anakinra has the ability of crossing the blood–brain barrier and reduces inflammation in the central nervous system (CNS) [207]. The reported difference in treatment responses to these two IL-1β pathway blocking agents may be due to different success in entering the CNS [208]. All these data underscore Anakinra as a promising treatment to reduce blood pressure in essential hypertension patients.

### 4.6. Gasdermin D Inhibitors

Although there is no direct proof in the literature that cell pyroptosis directly contributes to the pathological process of hypertension, this type of cell death could represent a promising pathway to reduce blood pressure. There are several compounds to inhibit GSDMD activity, and all of them directly target its cysteine (Cys) residues. The molecular structure of Cys contains a thiol group (Cys-SH) susceptible to oxidation by ROS, and thus, this residue serves an important structural role in many proteins. The oxidation of thiol groups results in sulfenic (Cys-SOH), sulfinic (Cys-SO_2_H), and sulfonic (Cys-SO_3_H) groups, which tend to create a covalent bond with other Cys residues to form disulfide bonds [209]. Disulfide bonds between Cys groups (Cys-SS-Cys) are crucial for protein folding and stability, can modify protein rigidity, and affect their proteolytic resistance [210]. Thus, disulfide bonds in proteins are formed consequently of the oxidation of cysteine thiol group residues [209] and a pro-oxidant environment could avoid their formation. Additionally, chemical modifications of the cysteine thiol group such as succinylation [211] could also prevent the formation of disulfide bonds. 

In this context, it has been demonstrated that the oxidation of the cysteine residues of GSDMD^NT^ is needed for pore formation, in particular, oxidation of Cys191 [177,212]. The α-Cys-reactive drug necrosulfonamide (NSA) inhibits pyroptosis in human and mouse cells by disrupting disulfide bonds formed at Cys191 [177]. Disulfiram is another potent inhibitor of GSDMD pore formation in vitro and in vivo, and covalently modifies human/mouse Cys191/Cys192 in GSDMD to block pore formation [178]. Moreover, dimethyl fumarate modifies GSDMD and GSDME through succinylation on Cys192 in mice, which is equivalent to Cys191 in human, and prevents its interaction with caspases, blocking its ability to induce pyroptosis [179]. The Cys sulfhydryl group is nucleophilic and easily oxidized by ROS production; therefore, antioxidant compounds may have an important role in GSDMD inhibition by preventing the oxidation of its Cys groups. Several studies demonstrate the ability of antioxidants to reduce IL-1β expression and to mitigate pyroptosis in renal cells. Some of these antioxidant compounds are salvianolic acid B (SalB) [180], EPZ015666 (a specific inhibitor protein arginine methyltransferase 5 (PRMT5)) [181], dihydromyricetin (DHM) [182], sulforaphane [183], parthenolide [184], neferine [185], and sodium butyrate [141]. Interestingly, some of these antioxidants (salvianolic acid B [186], dihydromyricetin [187], sulforaphane [188], neferine [189], and sodium butyrate [190]) have also been reported to reduce blood pressure in animal models, suggesting that part of their antihypertensive effects could be mediated through the inhibition of GSDMD. In addition, all these antioxidant compounds have in common their ability to stimulate nuclear factor erythroid 2-related factor 2 (Nrf2) pathways [180,181,182,213,214,215], a master regulator of antioxidant genes with the capability of attenuating inflammasome activity [216]. Our group reported that the Nrf2 inducer bardoxolone normalizes oxidative stress-dependent hypertension associated with the depletion of renal DJ-1 in mice [217].

Taken together, this evidence suggests that the potential actions of these compounds on blood pressure may be mediated by the attenuation of pyroptosis, and thus, the inflammasome and pyroptosis are promising pharmacological targets for the development of novel antihypertensive drugs. 

## 5. Conclusions and Future Perspectives

Growing evidence highlights the activation of the inflammasome and the consequent pyroptosis as emerging mediators of the low-grade inflammation that is typical of the hypertensive state. Macrophages, T cells, and dendritic cells have been extensively demonstrated to be crucial for the development and progression of hypertension. The inflammasome pathway is upstream of the activation of adaptive and innate immune cells, and pyroptosis is known to exacerbate the kidney and vascular dysfunction that worsens hypertension. Thus, the inflammasome pathway is rising as a promising target for the development of novel antihypertensive drugs. In fact, drugs that are already in the market for the treatment of inflammatory disorders, like Anakinra, as well as experimental NLRP3 inflammasome inhibitors like MCC950, demonstrate effective decrease in blood pressure in the experimental setting. In addition, blockers of GSDMD activity also show promise as potential tools for the attenuation of hypertension and hypertension-induced end-organ damage. The inflammasome/pyroptosis signal cascade is one of the most effective pathways to induce inflammation and is emerging as a promising target to develop novel therapeutic approaches to block inflammation and prevent hypertension.

## Figures and Tables

**Figure 1 ijms-22-01064-f001:**
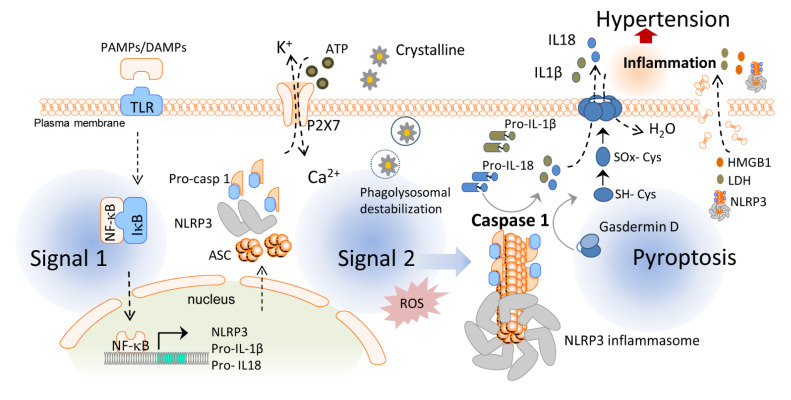
Activation of the inflammasome and pyroptosis induce hypertension. In the first step of the inflammasome activation, pathogen-associated molecular patterns (PAMPs) and damage-associated molecular patterns (DAMPs) stimulate toll-like receptors (TLR) and the translocation of nuclear factor kappa B (NF-κB) to the cell nucleus, which, in turn, increases the transcription of the nucleotide-binding domain, leucine-rich-containing (NLR) family pyrin domain containing 3 (NLRP3) inflammasome sensor, its posttranscriptional modification, and expression of pro-interleukin (IL)-1β and pro-IL-18. The second signal such as crystalline particles or P2X purinergic receptor 7 (P2 × 7) activation via ATP induces the oligomerization of the NLRP3 inflammasome complex which leads to the activation of caspase-1. Caspase-1 cleaves gasdermin D and converts pro-IL-1β and pro-IL-18 into mature IL-1β and IL-18. Pyroptosis occurs by the insertion of the N-terminal fragment of gasdermin D into the plasma membrane, creating oligomeric pores and allowing for the release of pro-inflammatory cytokines such as IL-1β and IL-18 to the extracellular space. Pore formation also induces water influx into the cell, cell swelling, and osmotic cell lysis which induce further inflammation and hypertension by releasing more inflammatory products from the intracellular space. HMGB1: high mobility group box 1; IκB: inhibitor of κB; LDH: lactate dehydrogenase.

**Figure 2 ijms-22-01064-f002:**
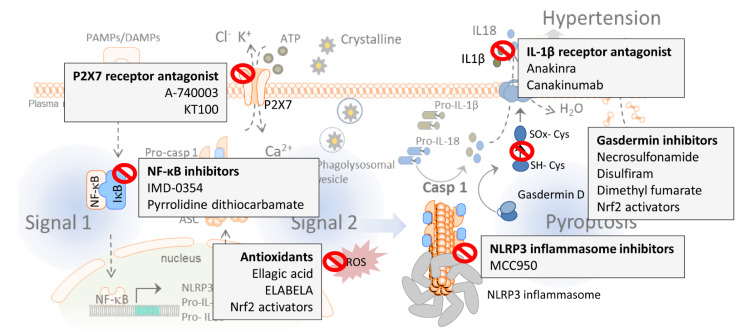
Possible pharmacological approaches targeting the nucleotide-binding domain, leucine-rich-containing (NLR) family pyrin domain containing 3 (NLRP3) inflammasome and pyroptosis pathways to reduce blood pressure.

**Table 1 ijms-22-01064-t001:** Potential pharmacological approaches targeting the NLRP3 inflammasome and pyroptosis to reduce blood pressure.

Drug	Mechanism of Action	Effect on Blood Pressure Regulation	Organism	Publications
A-740003	P2X7 receptor antagonist	Decreases mean RV pressure associated with pulmonary hypertension	Rat	[123]
PKT100	P2X7 receptor antagonist	Improves survival in the pulmonary hypertension model	Mouse	[172]
Anakinra	IL-1 receptor antagonist	Reverses salt-induced hypertension/reduces pulmonary blood pressure	Mouse/human	[172,173,174,175]
Canakinumab	IL-1β blocker	No effect on blood pressure regulation	Human	[176]
NSA	Gasdermin inhibitor	No evidence	Human/mouse	[177]
Disulfiram	Gasdermin inhibitor	No evidence	Human/mouse	[178]
DMF	Gasdermin inhibitor	No evidence	Mouse	[179]
Nrf2 activators	Gasdermin inhibitors, antioxidants	Reduces systolic blood pressure	Mouse	[180,181,182,183,184,185,186,187,188,189,190]
MCC950	NLRP3 inflammasome inhibitor	Reverses salt-induced hypertension/reduces blood pressure	Mouse/rat	[129,130,160,191,192]
Ellagic acid	Antioxidant	Reduces pulmonary artery hypertension	Rat	[121]
ELABELA	Antioxidant	Mitigates hypertension	Human renal tubular cells	[193]
IMD-0354	NF-κB inhibitor	Reduces RV pressure	Rat	[194]
PDTC	NF-κB inhibitor	Reverses salt-induced hypertension	Rat	[163]

DMF, dimethyl fumarate; NSA, necrosulfonamide; PDTC, pyrrolidine dithiocarbamate; RV, right ventricular.

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
