# Peer review of "Emerging Role of the Inflammasome and Pyroptosis in Hypertension"

_ijms, 2021, doi:10.3390/ijms22031064_

Round 1
Reviewer 1 Report
The manuscript entitled “Emerging role of the inflammasome and pyroptosis in hypertension” provides a comprehensive review of the relationship between inflammasome/pyroptosis and hypertension and it is quite useful for readers to understand the basic knowledge about this field. However, some parts of this review need to be improved and I hope my suggestions can help the author refine it.
Line 14-15, it is inappropriate to clarify that pyroptosis is a caspase-1-dependent type of cell death since caspase-4/11 could also mediate pyroptosis. (PMID: 26375003)
Why the author continuously emphasized that low-grade inflammation contributed to hypertension (line 45, line 83). Does high-grade inflammation have other effects?
The information from line 96-103 will mislead the readers that ASC is required for NLRP1 inflammasome activation. In fact, NLRP1 inflammasome can be activated without ASC (PMID: 24492532).
It is better to introduce NF-Kb signaling in the paragraph from line 145 since you mentioned a lot about the role of NF-Kb in hypertension later.
The author wanted to demonstrate the relationship between inflammasome and hypertension. However, the manuscript main focuses on the role of NLRP3 inflammasome in hypertension. Actually, other inflammasome genes, such as NLRC4, AIM2, are also involved in hypertension and vascular dysfunction (PMID: 28092664; PMID: 29439156). Thus, more thorough information is recommended to be added in this review.
A lot of novel concepts about the mechanisms of inflammasome activation are proposed in recent years (PMID: 32943500; PMID: 30487600; PMID: 33268895). I recommend the author to update the knowledge of that part and provide more recent findings in this field.
Besides, the format and grammar need to be checked.
Author Response
The manuscript entitled “Emerging role of the inflammasome and pyroptosis in hypertension” provides a comprehensive review of the relationship between inflammasome/pyroptosis and hypertension and it is quite useful for readers to understand the basic knowledge about this field. However, some parts of this review need to be improved and I hope my suggestions can help the author refine it.
Line 14-15, it is inappropriate to clarify that pyroptosis is a caspase-1-dependent type of cell death since caspase-4/11 could also mediate pyroptosis. (PMID: 26375003)
Response: The text has been corrected to address this comment.
Why the author continuously emphasized that low-grade inflammation contributed to hypertension (line 45, line 83). Does high-grade inflammation have other effects?
Response: Unlike acute inflammation where there is a drastic increase of the inflammatory response that then eventually subsides (i.e. infection), low grade inflammation is defined as chronic and persistent low levels of inflammation throughout the body. Hypertension is a chronic disease per definition and persistent, low grade inflammation (not acute) has been identified as an integral part in its pathogenesis. The molecular mechanisms and the consequences of acute and persistent inflammation are very different - this is the reason of emphasizing it in the manuscript, so the reader that is unfamiliar with hypertension does not confuse it with acute inflammation. We acknowledge that it may not have been clear in the first version of this manuscript and have now clarified that low grade inflammation is persistent inflammation the first time that it is used in the text.
The information from line 96-103 will mislead the readers that ASC is required for NLRP1 inflammasome activation. In fact, NLRP1 inflammasome can be activated without ASC (PMID: 24492532).
Response: We agree with the reviewer; this reference has been added to the manuscript and a comment has been included in the text.
It is better to introduce NF-Kb signaling in the paragraph from line 145 since you mentioned a lot about the role of NF-Kb in hypertension later.
Response: Thank you, a comment has been included in the text.
The author wanted to demonstrate the relationship between inflammasome and hypertension. However, the manuscript main focuses on the role of NLRP3 inflammasome in hypertension. Actually, other inflammasome genes, such as NLRC4, AIM2, are also involved in hypertension and vascular dysfunction (PMID: 28092664; PMID: 29439156). Thus, more thorough information is recommended to be added in this review.
Response: We agree with the reviewer that there are other inflammasomes that may be involved in hypertension; however, most of the papers in the hypertension literature have focused specifically on the NLRP3 inflammasome, thus the NLRP3-centered review. Yet, we mention the possible role of the NLRC4 inflammasome in blood pressure regulation (end of the section 3.2; page 12). Regarding AIM2, direct evidence of its involvement in the development or progression of hypertension is still lacking in the literature.
A lot of novel concepts about the mechanisms of inflammasome activation are proposed in recent years (PMID: 32943500; PMID: 30487600; PMID: 33268895). I recommend the author to update the knowledge of that part and provide more recent findings in this field.
Response: Response: All the mechanisms that have been described for the activation of the inflammasome would be worth of their own in-depth review paper. However, in the present review we chose to focus on the interplay between the inflammasome and pyroptosis with hypertension, and, due the journal character limit, we only review the most well-known and accepted activation mechanisms. However, per your suggestion, we included the references that you mentioned and refer the reader to those papers for further reading on those novel mechanisms of activation.
Besides, the format and grammar need to be checked.
Response: Per your request, the format and grammar have been re-checked.
Reviewer 2 Report
This is an excellent and vey comprehensive review on a timely and important topic. It is well written, and key information in the field are thoroughly summarized. The data are well discussed and put into context. The text is easy to read. The figures are excellent additions and are aesthetically pleasing. In general, the data are well discussed, and the organization of the review is logical (see some comment on this below).
There are a couple of suggestions and comments aimed at further improving the review:
1, The cause-effect relationship and the questions surrounding such determinations should be more explicitly discussed. Throughout the text it is not always clear whether inflammation and inflammasome activation is clearly proven to be the cause of hypertension, or if it can be merely considered as a result of it. Possible uncertainty in this respect, or strong proof for one way or the other should be more clearly stated. In fact, uncertainties of this aspect are most explicitly discussed towards the end of the review in section 4.5, where the authors state that the question still remains whether IL1beta and IL-18 are inflammatory markers or mediators of hypertension (line 209-210).
More explicit discussion on the issue of cause-effect relationship would be most beneficial in section 3 where the role of inflammasome in the kidney is discussed (starting at line 249). The kidneys can be both initiators of blood pressure elevation and targets of this conditions. In the latter case, hypertension can promote kidney injury and tubulointerstitial fibrosis. In this section it is not clear whether the different experiments and evidence discussed separates causal effects (most likely many of the studies do not). Therefore, it is equally possible that inflammasome activation is a result of hypertension leading to damage (which then may via a positive feed back augment hypertension). While the cause-effect in many cases can be difficult to determine, this question would be important to discuss.
2, Section 3 is somewhat long, subsections breaking it up would be beneficial.
3, The authors do not distinguish different forms of hypertension. The majority of the text deals with systemic hypertension, but the data are not grouped according to origin (e.g. essential, eclampsia-induced; renal hypertension etc). While organizing the text based on pathological origin likely would be extremely difficult due to the diversity of the models, wherever possible, the pathology in the individual studies should be clearly indicated. Most importantly though, various forms of systemic hypertension and pulmonary hypertension are discussed together in the same section. It would make more sense to seperate the discussion on pulmonary hypertension (line 196-212) either into a separate subsection, or at least discuss it at the end of section 3 and not to mix it with data on systemic hypertension, since these are clearly two different pathological entities.
- Consider referring to Figure 1 earlier in the section discussing inflammasomes.
Minor:
-The word “member” (“member of the innate immune response”) in the abstract (line 11) is somewhat awkward, please consider replacing (e.g. component of… or contributor to).
-The sentence starting at line 87 (“Inflammasomes are…”) is difficult to understand, please rephrase.
-Line 185: elevated levels of NFkB: do the authors mean elevated activation levels?
-Line 198: “inflammasome expression” Do the authors mean activation? If not, what exactly does expression in this case mean? Expression of components of inflammasome? Please clarify.
-Line 336: some CoX inhibitors have been also suggested to elevate blood pressure similar to NSAIDs.
-Line 385-6. The abbreviations CCL2 and VCAM-1 have been used earlier but are defined only here. Please define at first use.
-Line 406. Please indicate that these drugs are ACE inhibitors of ARBs. What is the mechanisms through these affect LPS-induced effects? A bit more explanation for this result would be beneficial.
Author Response
Reviewer 2
This is an excellent and very comprehensive review on a timely and important topic. It is well written, and key information in the field are thoroughly summarized. The data are well discussed and put into context. The text is easy to read. The figures are excellent additions and are aesthetically pleasing. In general, the data are well discussed, and the organization of the review is logical (see some comment on this below).
There are a couple of suggestions and comments aimed at further improving the review:
1, The cause-effect relationship and the questions surrounding such determinations should be more explicitly discussed. Throughout the text it is not always clear whether inflammation and inflammasome activation is clearly proven to be the cause of hypertension, or if it can be merely considered as a result of it. Possible uncertainty in this respect, or strong proof for one way or the other should be more clearly stated. In fact, uncertainties of this aspect are most explicitly discussed towards the end of the review in section 4.5, where the authors state that the question still remains whether IL1beta and IL-18 are inflammatory markers or mediators of hypertension (line 209-210).
More explicit discussion on the issue of cause-effect relationship would be most beneficial in section 3 where the role of inflammasome in the kidney is discussed (starting at line 249). The kidneys can be both initiators of blood pressure elevation and targets of this conditions. In the latter case, hypertension can promote kidney injury and tubulointerstitial fibrosis. In this section it is not clear whether the different experiments and evidence discussed separates causal effects (most likely many of the studies do not). Therefore, it is equally possible that inflammasome activation is a result of hypertension leading to damage (which then may via a positive feed back augment hypertension). While the cause-effect in many cases can be difficult to determine, this question would be important to discuss.
Response: It is true that the role of the inflammasome as inducer or consequence in the hypertensive setting is controversial and still an active topic of investigation in the field. Whenever possible and clear from the literature, we have included comments to this effect within the text.
2, Section 3 is somewhat long, subsections breaking it up would be beneficial.
Response: Thank you for this comment. Five sub-sections have been included under section 3.
3, The authors do not distinguish different forms of hypertension. The majority of the text deals with systemic hypertension, but the data are not grouped according to origin (e.g. essential, eclampsia-induced; renal hypertension etc). While organizing the text based on pathological origin likely would be extremely difficult due to the diversity of the models, wherever possible, the pathology in the individual studies should be clearly indicated. Most importantly though, various forms of systemic hypertension and pulmonary hypertension are discussed together in the same section. It would make more sense to seperate the discussion on pulmonary hypertension (line 196-212) either into a separate subsection, or at least discuss it at the end of section 3 and not to mix it with data on systemic hypertension, since these are clearly two different pathological entities.
Response: As you mentioned in your comment, organizing the evidence based on the pathological origin is challenging because of the great variety of models used by the different reports. However, in this revised version and as much as possible, we have indicated the kind of hypertension studied in each report that we mention.
- Consider referring to Figure 1 earlier in the section discussing inflammasomes.
Response: Done
Minor:
-The word “member” (“member of the innate immune response”) in the abstract (line 11) is somewhat awkward, please consider replacing (e.g. component of… or contributor to).
Response: Done
-The sentence starting at line 87 (“Inflammasomes are…”) is difficult to understand, please rephrase.
Response: Done
-Line 185: elevated levels of NFkB: do the authors mean elevated activation levels?
Response: Corrected
-Line 198: “inflammasome expression” Do the authors mean activation? If not, what exactly does expression in this case mean? Expression of components of inflammasome? Please clarify.
Response: Corrected
-Line 336: some CoX inhibitors have been also suggested to elevate blood pressure similar to NSAIDs.
Response: Yes, we have now acknowledged in the text this suggested effects of COX inhibitors.
Line 385-6. The abbreviations CCL2 and VCAM-1 have been used earlier but are defined only here. Please define at first use.
Response: The definitions have been added at first use.
-Line 406. Please indicate that these drugs are ACE inhibitors of ARBs. What is the mechanisms through these affect LPS-induced effects? A bit more explanation for this result would be beneficial.
Response: Additional comments have been included in the text.
Reviewer 3 Report
This is a very well written review article, which addresses an interesting topic describing the roles of NLRP3 inflammasome and pyroptosis in hypertension. The article cites primarily recent scientific findings, which are nicely organized in four sub-headings and cover the most problems in the field. I have no concerns and would like just to suggest to convert Figure 2 into a Table including appropriate references to a study and if a pharmacological approach was tested in animal models and/or humans. However, this is just a suggestion and the article can stand as it is.
Author Response
This is a very well written review article, which addresses an interesting topic describing the roles of NLRP3 inflammasome and pyroptosis in hypertension. The article cites primarily recent scientific findings, which are nicely organized in four sub-headings and cover the most problems in the field. I have no concerns and would like just to suggest to convert Figure 2 into a Table including appropriate references to a study and if a pharmacological approach was tested in animal models and/or humans. However, this is just a suggestion and the article can stand as it is.
Response: Thank you so much for this suggestion, a table has been now included.
Reviewer 4 Report
This is an excellent review on the role of pyroptosis and its relationship to hypertension. The authors summarize what is currently know about the subject and have presented figures describing the role of pyroptosis.
I have no comments to add to this article.
Author Response
This is an excellent review on the role of pyroptosis and its relationship to hypertension. The authors summarize what is currently know about the subject and have presented figures describing the role of pyroptosis.
I have no comments to add to this article.
Response: Thank you so much for your time reviewing our manuscript.